# Kinds of Harm: Animal Law Language from a Scientific Perspective

**DOI:** 10.3390/ani12050557

**Published:** 2022-02-23

**Authors:** Tomasz Pietrzykowski, Katarzyna Smilowska

**Affiliations:** 1Research Centre for Public Policy and Regulatory Governance, Faculty of Law and Administration, University of Silesia, 40-007 Katowice, Poland; 2Department of Neurology, Regional Specialist Hospital im. Św. Barbary, 41-200 Sosnowiec, Poland; kasia.smilowska@gmail.com

**Keywords:** law, language, pain, suffering, distress, harm

## Abstract

**Simple Summary:**

Clarity and consistency of legal language are essential qualities of the law. Without a sufficient level of those, legal acts are often hardly capable of determining legal duties appropriately. The review of rules governing the protection of animals reveals that the current language remains far from satisfactory. Therefore, the paper discusses the most sensitive notions relevant for defining animal harm and argues for action to make the legal language of respective legal regulations more scientifically accurate and adequate to the actual needs of animal protection.

**Abstract:**

The current language of animal welfare laws is inconsistent and unclear in the basic terms pertaining to animal sensations and potential harms. In the case of law, the exact language often plays an essential role in determining legal duties and establishing their scope. Thus, for further progress in animal law, a coherent and unified basic conceptual framework is needed. To establish such a framework, the relevant legal terminology should be discussed in a prescriptive rather than interpretive manner while drawing on the medical sciences. Here, we propose a way to clarify the relevant terms to prevent misconceptions and improve the consistency of animal law.

## 1. Introduction

A review of the terminology appearing in animal welfare legislation across many countries indicates that there is no well-established, uniform or commonly accepted conceptual apparatus of the legal language relating to animal sensations, nor are there agreed-upon definitions of key terms. Terms such as ‘suffering’, ‘pain’, ‘distress’, ‘anxiety’, ‘fear’ and ‘harm’ are the most commonly used within the different European and American welfare laws. Inconsistency in their use and the lack of clarity regarding the relations among respective terms may hinder the development of coherent foundations for the legal protection of animals. Even if the interpretation of law is often able to overcome the inconsistencies of the language used by the legislator and improve the coherence and clarity of the conceptual apparatus of the law, the manner in which the terms are used in the legal texts is by no means immaterial. Its letter is always the point of departure and reference for interpretation and is often used as a criterium of the interpretive correctness.

Several examples may serve to illustrate the inconsistency in legal acts. The UK Animal Welfare Act uses the concept of suffering and seemingly equates it with distress (Art. 18), defining it as ‘physical or mental suffering’. In contrast, the US Animal Welfare Act does not contain any definition, and instead uses the concept of suffering as an alternative to distress (e.g., referring to ‘alleviating pain or distress’) or hurting (‘suffering or hurting’). In comparison, the German Animal Protection Act (2004) quite consistently uses the term ‘pain, suffering or harm’ (*Schmerzen, Leiden, Schäden*). Similar terms are proposed in Swiss law, which, however, replaces the word ‘harm’ with the word ‘fear’ in many cases (*Schmerzen, Leiden oder Ängsten*). In the Netherlands, pain, injury, physical and physiological discomfort, fear and chronic stress are mentioned separately. As a rule, the relatively new Swedish law (2018) uses only the concept of suffering, which in some cases is supplemented with ‘diseases’, as an alternative to ‘hurt’ or ‘discomfort’.

The European Union (EU) Directive 63/2010 on the protection of animals used for scientific or educational purposes declares that it has laid down rules for the protection of animals due to their ‘capacity to sense and express pain, suffering, distress and lasting harm’ (OJ L 276). EU Regulation 1009/2009 on the protection of animals at the time of killing allows, in turn, for animals to be induced to experience ‘pain, distress, fear or other forms of suffering’ (OJ L 303). The European Convention for the Protection of Animals for Slaughter from 1979 established that its provisions were designed to ‘spare animals suffering and pain’ (CETS no 102), while the European Convention for the Protection of Animals kept for Farming Purposes from 1976 stipulated that inflicting ‘unnecessary suffering or injury’ of animals should be avoided (CETS No 82). Still another term can be found in the EU Regulation on Trade in Seal Products 1007/2009, in which seals were recognised as ‘sentient beings that can experience pain, distress, fear and other forms of suffering’ (OJ L 286).

It is easy to get the impression of a major inconsistency bordering on chaos given the different meanings assigned to individual terms and the way in which their interrelations should be comprehended. In the case of law, this can be dangerous, since the text and its precision may often play an essential role in determining legal duties and establishing their scope. Thus, a coherent and broadly accepted conceptual framework for animal law must be worked out. It may be extremely helpful for lawyers that have to interpret and apply the legal rules properly as long as the latter remain laid down in such an unclear and incoherent terminology as the animal laws employ today. Ultimately, it is even more important for the legislators themselves so that the relevant laws may become better-drafted and therefore more coherent and scientifically-informed basis for animal handling. To accomplish this, the correct legal terminology should be discussed in a prescriptive manner by drawing on the medical sciences (i.e., physiology, neurobiology and psychology). Irrespective of the intricacies and traps of the law-making process partially determined by political considerations, the scientific and linguistic precision of drafting bills pertaining to animal law is a value in itself.

## 2. Pain and Distress

Pain, distress and suffering are terms usually applied to humans to describe states of the mind, such as perceptions or experiences [1,2]. Even though these definitions were established to improve the recognition of symptoms in clinical practice, there are still controversies regarding their applicability. Nonetheless, the growing body of scientific evidence on the animals well-being, entails the need to identify and understand relevant animal feelings.

The classical definition of pain includes the elements of pain perception and tissue damage. Importantly, verbal description is only one of several possible behaviours that express pain, and the inability of verbal communication should not negate the possibility that a human or a non-human animal can experience pain [3,4]. Thus, further advances in the field of animal pain include psychophysical and phenomenological studies with a consideration of verbal limitations (in accurately describing the experience of pain). In this regard, it has been shown that the experience of mild to moderate pain does not always lead to verbal and behavioural manifestations (despite physiological processes in the nervous system) even though it can be observed with further advances in the field.

Sensory neurons, which are located in most structures of the body, including the skin, muscle, joints and viscera, are called nociceptors or the so-called pain receptors. [5,6]. The intensity and duration of pain sensation are determined by the different types of nociceptors. Activation of cutaneous nociceptors can be caused by a wide range of adequate stimuli, including mechanical, thermal or chemical stimuli, as well as by inflammation in the area of the surface membrane receptors [5]. Activation of visceral nociceptors is mediated through stretching and the information proceeds through central pathways, including afferent nerve fibres from the tissues to the spinal cord (and medullary) dorsal horn via the ascending nerve tracts. Intraneuronal networks of the dorsal horn transmit nociceptive information to neurons projecting to the brain. Additionally, pain can also be accompanied by motivational-behavioural and autonomic responses. This includes increased heart rate, blood pressure, endocrine changes, increased attention, arousal, anxiety, avoidance and suffering [7]. Most importantly, pain may trigger behavioural changes, making it recognisable to an external observer.

In contrast to pain, which is relatively well-defined, distress has many accepted definitions. In general, distress is characterised as an aversive, negative state in which coping- and adaptation processes fail to return an organism to physiological and/or psychological homeostasis [8,9,10]. In this regard, distress is a form of stress triggered by severe or chronic stressors or the accumulation of stressors and there is a progression from a state of copable stress to distress that can overwhelm the normal homeostatic mechanisms of the body [11]. This transition depends on the ability of the animal to control its normal emotions and also its ability to control its environment. Of note, based on the studies on mammals, an exposure to acute stress can reduce pain due to stress-induced analgesia [12]. Recently, this phenomenon was also reported in zebrafish [13]. In contrast, chronic stress leading to distress has been the background for studies on so-called learned helplessness [14]. These studies examined animal behaviour, specifically escape learning, using exposure to uncontrollable stressor, typically in the form of inescapable foot shocks, that was repeated in the same or similar conditions. The results confirmed that animals in such inescapable conditions (1) develop clear signs of distress, including vocalisation or gastric ulcers [15] and (2) the stress response can lead to changes in behaviour and in physiology [10,16]. Further studies on humans have linked learned helplessness with depression [14].

Distress relates to one of the functions of the central nervous system, namely the preservation of body homeostasis, both directly and indirectly, in response to stress. As such, the first response to any stressor is the immediate activation of the hypothalamic–pituitary–adrenal (HPA) axis with the release of specific hormones (e.g., neuropeptides and glucocorticoids). Chronic stress can cause maladaptive responses to stress and lead to suppression or dysregulation of HPA. This can result in heart disease, stomach ulcers, sleep disorders and psychiatric complications (e.g., depression or anxiety). In non-human animals, distress can manifest with subclinical pathological changes (e.g., hypertension and immunosuppression), which are obviously more difficult to recognise than usual maladaptive behaviours such as abnormal feeding or aggression. In this regard, distress has negative impact on multiple body functions. This is known as the ‘biological cost of distress’ and usually takes the form of a prolonged recovery period to reset body homeostasis [8,17].

## 3. Suffering

The most difficult concept to define is suffering. It is deeply philosophically loaded and commonly used in colloquial (not only legal) language in relation to both human and non-human animals. As such, it exceeds the limits of the strictly defined terminology of the medical or biological sciences. Based on a general understanding, suffering may be defined as a negative experience; it should be considered individually and treated as a potential threat to a person’s integrity [18]. Suffering in the organisms with the nervous systems sufficiently developed to experience it can result from primitive feelings, including hunger, thirst, heat, cold, pain, fear and exhaustion, as well as from higher-order feelings, such as frustration, boredom, loneliness and depression [18,19]. In this regard, suffering is linked with physical pain. As such, suffering should be considered as a broader concept, which includes pain, distress and all other negative emotional states of an intensity that overwhelms individual adaptive and coping mechanisms (depending on the species and other individual and environmental factors). Thus, suffering can be distinguishable into ‘suffering from pain’, ‘suffering from hunger’, ‘suffering from loneliness’ and so on.

Animals are obviously incapable of verbally expressing pain, distress or suffering. However, these states can be empirically assessed through observation of animal behaviours, which are displayed in avoidance reactions (e.g., escaping or moving away from stimuli). It is worth noting that similar observations can be made of humans who are unable to verbally express their feelings (e.g., infants or patients with aphasia and other disabilities limiting the possibility to communicate verbally).

The most reasonable approach to assess animal depositions is to recognise the associated clinical signs. The observation of behavioural reactions (e.g., escape, vocalisation or aggression) and physiological reactions (e.g., increase in blood pressure, temperature, respiration and levels of specific hormones such as adrenaline, prolactin, insulin or glucocorticoids) remain essential. Distress can also manifest as self-mutilation, weight loss, reduced activity, sleep loss, irritability and decreased mating and reproductive performance, as well as changes in urinary and bowel activities and a lack of grooming [20].

Additionally, increased levels of the so-called stress hormones, particularly cortisol, increase serum glucose levels and enhance the brain’s use of glucose, which can be easily detected in the serum. However, hormone concentration is more useful for detecting acute stress because concentrations of catecholamines may return to normal the range in response to persistent stressors. In contrast, faecal corticosteroids can be used as a marker of prolonged and chronic stress [21,22].

As a result, biochemical markers should be used in conjunction with behavioural and environmental data to analyse animal welfare. Furthermore, advances in neuroimaging, as an example of a non-invasive technique, can also be used to assess pain, distress and suffering. Neuroimaging in animals, including functional magnetic resonance imaging (fMRI), manganese-enhanced MRI, positron emission tomography and electroencephalography, allows us to better understand the mechanisms of pain [23].

In this view, suffering should be treated as denoting any negative emotional state that can be determined by means of detecting neurological or behavioural manifestations of pain, stress, non-trivial negative emotional state as well as the processes and stimuli that may cause them. Thus, suffering should not be regarded as an alternative to pain or distress. Both of them are forms of suffering. Furthermore, these two terms (pain and distress) have separate referents, although they may to some extent overlap. Pain in the sense of a conscious sensation is caused by a signal from the nociceptor, and distress is an unpleasant sensation caused by the body’s inability to restore balance; they may either coexist (with the same cause) or appear independently of each other. However, suffering should not be considered as a separate type of experience; rather, it should be understood as an overarching category that encompasses both pain and distress.

Other terms usually associated with animal rights, such as injury, discomfort, anxiety and fear, may be considered as possible experiences of pain, stress, distress or suffering. Injury is the damage of the body, and as such, it leads to pain. Discomfort, according to standard definitions is understood as a feeling of being physically or mentally uncomfortable and, accordingly, depending on its intensity or duration can become a suffering.

## 4. Conclusions: Harm and Its Kinds

At the same time, the concept of harm, which is also used in legal language in the context of animal protection, seems to have an even broader meaning that is categorically different. As opposed to pain, suffering, distress or injury it is predominantly legal rather than scientific term. It has been in wide use mainly in private law where it typically denotes the detriment to one’s interests that may be legal remedied. In case of animals, harm should also pertain to detriment of animal interests relevant to the law. A harm so conceived can of course consist of inflicting suffering on an animal by causing it to experience pain (especially from injury), distress or other negative sensations. However, an animal may be also harmed by worsening its situation, that is increasing the probability of their occurrence or reducing the probability of animal’s experiencing positive sensations (in particular by causing lasting deprivation of some capabilities—as in case of, e.g., blindness or lameness). Harming an animal would therefore consist of causing its actual or potential suffering, or depriving it of the benefit of achievable sensations of a positive nature. Thus, harm would constitute a detriment to the present or only probable quality of life of an animal. As in case of general conception of harm, for animals it may also take the form of a real damage to its interests (damnum emergens) or just a loss of potential future gains (lucrum cessans), albeit in entirely non-monetary domain.

To conclude, a uniform conceptual framework of animal welfare law should include suffering as an umbrella term for pain, distress and other kinds of unpleasant sensations. They are concepts having their scientific meaning and should be used in the legal language correspondingly. Animal law partially emerges from the scientific elucidation of the animal sentience and kinds of sensations it is capable to experience. It is therefore important to make the conceptual apparatus of the law as accurate and embedded in the relevant scientific knowledge as possible.

As opposed to that, the concept of harm belongs to the different realm of the legal terms developed to regulate the protection of human interests and remedies to their infringements. As such it may and should be adapted to the animal law too. Its main role should be to link the correctly used, scientifically informed terms referring to particular kinds of animal interests that may be negatively affected by human conduct with the legal qualification as detriment relevant for legal reaction. Harm, therefore, should not be regarded as an alternative or complement to the terms referring to particular experiences that may be its instantiations. It is rather a legally relevant negative state of affairs that may consist of pain, distress or another kind of animal suffering, increase their probability or decrease the probability of experiencing opposite, positive sensations.

This commentary is a call for action. Therefore, we regard that an interdisciplinary task force should be established to work out and propose a standardised, scientifically informed vocabulary for a legal use. This could substantially improve the quality of legislation and the practice of animal law.

## Data Availability

Not applicable.

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
