# Peer review of "Kinds of Harm: Animal Law Language from a Scientific Perspective"

_animals, 2022, doi:10.3390/ani12050557_

Round 1

Reviewer 1 Report

The paper analyses the significance of animal law language, concerning animal sensations. It criticizes the inconsistency of terminology and suggests an improvement of clarity and harmonization of terms concerning concepts such as pain, distress, suffering and harm. This review does not concern the definition of these terms from the point of view of “hard sciences”. As to the legal perspective, the article is well-structures and uses a clear and accurate language; it adopts an original perspective. It makes reference to some international and comparative legal documents and the conclusions are consistent with the premises.

  • Suggestions:
  • The EU acts (directive and regulation) mentioned at lines 45 concern specific fields (animal testing and animal protection at the time of the killing): they should be mentioned.
  • The authors might specify if the definitions’ harmonization they suggest should be adopted in different fields (e.g. farming, animal testing, animals’ transport etc.) or not. Besides, it would be interesting to identify the international organizations where the harmonization might be discussed.

Author Response

We thank the reviewer for this thoughtful comment. We have now adjusted the text on the page 2, line 50-55:

“The European Union (EU) Directive 63/2010 on the protection of animals used for scientific or educational purposes declares that it has laid down rules for the protection of animals due to their ‘capacity to sense and express pain, suffering, distress and lasting harm’ (OJ L 276). EU Regulation 1009/2009 on the protection of animals in the time of killing allows, in turn, for animals to be induced to experience ‘pain, distress, fear or other forms of suffering’ (OJ L 303)”.

Reviewer 2 Report

Extremely interesting analysis of a well defined context. I would really suggest to include in the conclusions a sort of proposal on how to realize the harmonization that is needed (i.e. the creation of a working group to write down guidelines....)

Author Response

We thank the reviewer for this thoughtful comment. We have added the following concluding remark on page 5, line 236-239-15.

“This commentary is a call for action. Therefore, we regard that an interdisciplinary task force should be established to work out and propose a standardized, scientifically informed vocabulary for a legal use. This could substantially improve the quality of legislation and the practice of animal law”.

Reviewer 3 Report

Important point, the transition between animal science and the law is difficult. Good exploration of legal words seen through the eyes of scientific information. But since most legislators are not scientist, it will be difficult to change long existing legislative language.  

Author Response

We thank the reviewer for this insightful reservation. This is part of the reason why we point out to the need of an interdisciplinary, scientific task force to work out and propose a unified, scientifically-informed vocabulary instead of natural language of legal drafters backed by their intuitions only. We have added the following concluding remark on page 5, line 236-239-15.

“This commentary is a call for action. Therefore, we regard that an interdisciplinary task force should be established to work out and propose a standardized, scientifically informed vocabulary for a legal use. This could substantially improve the quality of legislation and the practice of animal law”.

Reviewer 4 Report

The authors present a considered discussion of the terminology used in legislation around the concepts of animal welfare and associated ideas such as suffering and harm. The discussion is very much overdue and the manuscript offers a valuable and insightful injection into the literature of well-reasoned and scientifically-based approaches to animal welfare legislation and terminology. The authors present a coherent argument for the use of specific terms and, while some of the supporting evidence could be bolstered through further citations and examples from the existing scientific literature, the overall presentation is sound.

The one notable flaw in the manuscript is the lack of clear and emphatic definitions. As noted in one of the comments on the document itself, providing the readers with a clear and obvious definition for a given term (e.g. suffering) would go a long way towards providing the tools which legislators and scientists alike could use in a constructive and effective manner. Ironically, by failing to provide clear and concise definitions of the terms the authors perpetuate the very ambiguity which they identify as being the major shortcoming of animal welfare legislation. In a similar vein, it would be advisable for the authors to consider expanding their discussions of some of the concepts and possibly consider providing some examples of their application.

With this said, the authors should be commended on compiling an exceptional manuscript which is both important and timely and will undoubtedly prove useful in the legal and scientific arenas.

Author Response

We thank the reviewer for this useful comment. We agree that what is needed to improve the quality of legislation and the practice of animal law is providing clear and scientifically adequate definitions of the relevant terms as well as explaining the relations they stand in. However, we believe that such definitions to be relevant for law-making purposes should be provided by a special, interdisciplinary task force systematically reviewing the ambiguities and scientific nuances of all relevant terms. Our goal is much more modest – we focus on pointing out that there is an unmet need with regard of the consistency and scientific accuracy of the legal vocabulary being applied throughout legal systems. We claim that the problem is not local but rather reflect the general state of art, in which the legal drafters select the terms of key importance rather accidentally and on the basis of their lay intuitions mainly. Furthermore, we try to indicate the general way out by examining the scientific background of several notions and showing their potential relations. Nonetheless, the paper is a call for action rather than a definitive solution of the problems raised and explained therein.

Reviewer 5 Report

This paper may make a lot of good points for lawyers but there are several unsubstantiated claims as well as some inaccuracies, and the language is imprecise in many places.  

I have the following comments:

Line 61:           “It is important to interpret and apply the law properly– as long as it has to be done on the basis of the pieces of legislation with such an unclear and incoherent terminology as the animal laws employ today.”

Rewording: I suggest the authors mean that it is DIFFICULT AND important to interpret the law….  Furthermore, should they refer to the scientific basis as being of utmost importance, as otherwise the law itself will be challenged on the basis of poor or incorrect scientific data.

Line 69: precision rather than quality?

Line 76: use ‘feelings’ rather than dispositions

Line 81: Suggest: “Thus, further advances in the field of animal pain include psychophysical ….

Line 85: I would argue that moderate and often mild pain do lead to behavioural responses and sometimes, depending on the species, vocalisation (there are often marked differences for example between predators and predatees (prey animals).

Line 88: The quality of the pain : do you means intensity or the type of pain e.g. ache, stabbing, visceral, and/or its duration?

Line 92: activation of ‘somatic’ receptors as opposed to visceral receptors which take a different pathway via the sympathetic nervous system.

Line 97:  and most important for recognition of pain by an observer, behavioural changes

Line 104 stressful events

Line 105/108: stress is not an emotional state it is a stimulus that can lead to distress

Line 112: stressor not stress

Line 115: Suggest: can lead to changes in behaviour and in physiology

Line 137 et seq:  do you not think that suffering is the mental impact of all adverse states.  For example, Pain itself may be of no consequence unless an animal finds it aversive.  You go on to elucidate this in the next few lines.

Line 146: Animals are obviously incapable of verbally expressing pain, distress or suffering, but they can and do vocalise: It is useful to remember the quotation of Jeremy Bentham The question is not, Can they reason?, nor Can they talk? but, Can they suffer? Why should the law refuse its protection to any sensitive being?” (1789) 

Line 152:The authors might like to read Morton and Griffiths (1985) Guidelines on the recognition of pain, distress and discomfort in experimental animals and an hypothesis for assessment. Veterinary Record 116, 431-436.

Lines 160-163: True for the catecholamines but not for the corticosteroids which can last for longer depending on the length of time an animal is exposed to the stressor and faecal corticosteroids are used as a measure of longer term distress.

Lines 165 et seq: need a reference for this paragraph

Lines 178-179:please find a solid reference that pain and distress do not always have a mental component (i.e. some level of suffering).  That is P & D may be intertwined: pain always causes some level of distress, but distress does not usually cause pain.  However, in humans we talk about the pain of separation and grief but not in animals where pain feelings are usually restricted to the nociception system, and distress to a mental state.

Line 186: surely the level of suffering is not an all or none phenomenon, there are varying degrees.

Lines 197-198: Harming an animal would therefore consist of causing its actual or potential suffering, or depriving it of the benefit of achievable sensations of a positive nature. This latter points is referred to in EU legislation as ‘lasting harm’ e.g. blindness.

Author Response

Line 61: “It is important to interpret and apply the law properly– as long as it has to be done on the basis of the pieces of legislation with such an unclear and incoherent terminology as the animal laws employ today.”

We thank the reviewer for this comment. We have now adjusted the text appropriately. We hope that the reviewer will find this adjustment acceptable.

Rewording: I suggest the authors mean that it is DIFFICULT AND important to interpret the law….  Furthermore, should they refer to the scientific basis as being of utmost importance, as otherwise the law itself will be challenged on the basis of poor or incorrect scientific data.

We thank the reviewer for this comment. We have now adjusted the text accordingly.

Line 69: precision rather than quality?

We thank the reviewer for this comment. We have now adjusted the text.

Line 76: use ‘feelings’ rather than dispositions.

Thank you for this comment – we propose “sensations” rather than “feelings”.

Line 81: Suggest: “Thus, further advances in the field of animal pain include psychophysical ….

We thank the reviewer for this comment. We have now adjusted the text.

Line 85: I would argue that moderate and often mild pain do lead to behavioural responses and sometimes, depending on the species, vocalisation (there are often marked differences for example between predators and predatees (prey animals).

We thank the reviewer for this comment. Further in the same paragraph we stated, however, that further advances in the field proved that mild to moderate pain may be experienced even though behavioural responses can be limited. We have now adjusted the text on the page 2, line 80-85:

“Thus, further advances in the field include psychophysical and phenomenological studies with a consideration of verbal limitations (inaccurately describing the experience of pain). In this regard, it has been shown that the experience of mild to moderate pain does not always lead to verbal and behavioural manifestations (despite physiological processes in the nervous system) even though it can be observed with further advanced in the field”

Line 88: The quality of the pain : do you means intensity or the type of pain e.g. ache, stabbing, visceral, and/or its duration?

We thank the reviewer for this comment. We have now adjusted the text.

Line 92: activation of ‘somatic’ receptors as opposed to visceral receptors which take a different pathway via the sympathetic nervous system.

Thank you for your valuable comment. Our aim in this paragraph was to delineate pain related receptors and pathways excluding autonomous nerve system.

Line 97:  and most important for recognition of pain by an observer, behavioural changes

We thank the reviewer for this comment. We have now adjusted the text.

Line 104 stressful events

We thank the reviewer for this comment. We have now adjusted the text.

Line 105/108: stress is not an emotional state it is a stimulus that can lead to distress

Thank you for this comment– we understand stress as relatively short and inconsequential form of an aversive state, not a stimulus leading to distress. After correcting “stress” into “stressors” we hope our reasoning is clear now.

Line 112: stressor not stress

We thank the reviewer for this comment. We have now adjusted the text.

Line 115: Suggest: can lead to changes in behaviour and in physiology

We thank the reviewer for this comment. We have now adjusted the text.

Line 137 et seq:  do you not think that suffering is the mental impact of all adverse states.  For example, Pain itself may be of no consequence unless an animal finds it aversive.  You go on to elucidate this in the next few lines.

We agree that this interpretation. Although we assume that pain (as nociceptive signal) gives rise to an aversive mental state (suffering) in all organisms that have nervous systems developed enough to become sentient (that is capable of having subjective mental states of the most elementary nature). Therefore, as a matter of principle, pain leads to suffering (may be conceived as a kind of the latter) even if – in some exceptional, individual cases, pain might be experienced as a neutral or even pleasant sensation. Nonetheless, from an evolutionary perspective, the nociceptive signals developed to make organisms aversive to the types of incentives that may be dangerous to the organisms survival.

Line 146: Animals are obviously incapable of verbally expressing pain, distress or suffering, but they can and do vocalise: It is useful to remember the quotation of Jeremy Bentham The question is not, Can they reason?, nor Can they talk? but, Can they suffer? Why should the law refuse its protection to any sensitive being?” (1789) 

Thank you – we fully agree (even if not every case of pain may result in vocalisation – as e.g. in fish).

Line 152:The authors might like to read Morton and Griffiths (1985) Guidelines on the recognition of pain, distress and discomfort in experimental animals and an hypothesis for assessment. Veterinary Record 116, 431-436.

Thank you very much for this comment. We are aware of this publication. It it’s definitely very helpful and important literature even thought it’s not very recent.  We added it to the reference list.

Lines 160-163: True for the catecholamines but not for the corticosteroids which can last for longer depending on the length of time an animal is exposed to the stressor and faecal corticosteroids are used as a measure of longer term distress.

We thank the reviewer for this useful comment, it is good to further specify. The aim of this paper is mainly to show neuroscience perspective in the legal angle. Therefore, we tried to limit complex neuroscience background. Cortisol as a stress hormone is a marker of acute stress in serum, but as you thoughtfully comment in can also serve as a marker of prolonged and chronic stress in different biological material. We have now changed this paragraph on page 4, line 172-177:

“However, hormone concentration is more useful for detecting acute stress because concentrations of catecholamines may return to normal the range in response to persistent stressors. In contrast, faecal corticosteroids can also be used as a marker of prolonged and chronic stress”. 

Lines 165 et seq: need a reference for this paragraph

We thank the reviewer for this comment. We have now adjusted the text.

Lines 178-179: please find a solid reference that pain and distress do not always have a mental component (i.e. some level of suffering).  That is P & D may be intertwined: pain always causes some level of distress, but distress does not usually cause pain.  However, in humans we talk about the pain of separation and grief but not in animals where pain feelings are usually restricted to the nociception system, and distress to a mental state.

Thank you for your comment. We fully agree, however, distress doesn’t have to cause pain (in the nociceptive sense). „Pain” is also sometimes used in broader meaning to cover not only physical stimuli triggering nociception but rather psychological distress (like in pain of separation) which is an additional source of conceptual ambiguities. Therefore, we are not sure what change in the text the reviewer finds necessary.

Line 186: surely the level of suffering is not an all or none phenomenon, there are varying degrees.

We thank the reviewer for this comment we fully agree.

Lines 197-198: Harming an animal would therefore consist of causing its actual or potential suffering, or depriving it of the benefit of achievable sensations of a positive nature. This latter points is referred to in EU legislation as ‘lasting harm’ e.g. blindness.

We thank the reviewer for this comment. We have now adjusted the text on page 4, line 191-195:

“However, an animal may be harmed also by worsening its situation, that is increasing the probability of their occurrence or reducing the probability of animal’s experiencing positive sensations (in particular by causing lasting deprivation of some capabilities – as in case of e.g. blindness or lameness)”.

Round 2

Author Response

Please find the reply attached 
